# Using a Gender-Responsive Land Rights Framework to Assess Youth Land Rights in Rural Liberia

**Elizabeth Louis [1,*], Tizai Mauto [1], My-Lan Dodd [1], Tasha Heidenrich [1], Peter Dolo [2] and Emmanuel Urey [1]**

[1] Landesa, 1424 4th Ave, Seattle, WA 98101, USA; tizaim@landesa.org (T.M.); myland@landesa.org (M.-L.D.); tashaheid295@gmail.com (T.H.); emmanuelu@landesa.org (E.U.)

[2] Development Education Network-Liberia (DEN-L), Dementa Road, Gbarnga, Bong Country, Liberia; dev_edunet@justemail.net

[*] Correspondence: elizabethl@landesa.org

**Abstract:** This article summarizes the evidence on youth land rights in Liberia from a literature review combined with primary research from two separate studies: (1) A qualitative assessment conducted as formative research to inform the design of the Land Rights and Sustainable Development (LRSD) project for Landesa and its partners' community level interventions; and (2) a quantitative baseline survey of program beneficiaries as part of an evaluation of the LRSD project. The findings are presented using a Gender-Responsive Land Rights Framework that examines youth land rights through a gender lens. The evidence highlights that female and male youth in Liberia face significant but different barriers to long-term access to land, as well as to participation in decisions related to land. Our suggested recommendations offer insights for the implementation of Liberia's recently passed Land Rights Act as well as for community-level interventions focused on increasing youth land tenure security in Liberia.

**Keywords:** youth land rights; gender-responsive land rights framework; Liberia Land Rights Act; land governance; tenure security

## 1. Introduction

Broad-based land tenure security and equitable land governance are pressing issues in Liberia. Land and natural resources have always been, and remain, crucial to Liberia's economy [1,2]. Seventy percent of the active population is dependent on agriculture for their livelihood and over half of the country's inhabitants live in rural areas [3]. Youth (ages 15–35) constitute approximately 34% of the population [1] and rural youth depend primarily on agriculture to support their livelihoods [4]. Many rural youth lack access to farmland or suffer from high levels of land tenure insecurity.

The viability of the youth demographic is crucial to Liberia's social, political, and economic future. The exclusion of youth from effectively accessing land to support themselves and their families could have negative implications for the transfer of knowledge and skills, as well as food security, youth employment, economic development, and security. The civil war arose from the systematic denial of land (and other economic assets) and exclusion from governance of the indigenous Liberians who constitute the majority of Liberia's population [5]. After the conflict, the country made strides

---

[1] Although this age range is generally accepted as a definition of youth, in reality there are wide range of social factors that define youth and adulthood in rural communities in Liberia. The Government of Liberia in its National Youth Policy defines Liberian youth as being between the ages of 15 and 35 (Brownlee et al., 2012), a definition also used by the African Youth Charter (2006) and the National Federation of Liberian Youth (Nasser 2012).

toward peace, stability, and economic growth. However, poverty, food insecurity, inadequate human capacity and infrastructure, a high unemployment rate, particularly among youth, and land tenure insecurity threaten further progress.

The Land Rights Act, adopted in September 2018, was formulated to address several inequities in land access and land governance, giving communities ownership rights and empowering them to make decisions on the lands that they have customarily accessed for decades [6]. The LRA explicitly recognizes the rights of all community members so long as they meet the LRA's legal definition of community member. Through these provisions, women, youth, and "strangers" [2]—groups that have been traditionally marginalized within rural communities—enjoy land rights by operation of the LRA. Landesa worked with the Government of Liberia to provide technical input into the framing of the Land Rights Act. Additionally, through the Land Rights and Sustainable Development Program (LRSD), Landesa currently works with local CSO partners DEN-L and FCI to raise awareness around land rights for women and youth in order to foster sustainable development in rural communities.

This paper summarizes the evidence from a literature review combined with primary research from two separate studies conducted by Landesa, which focused on youth land rights in Liberia. The particular issues that youth face with respect to land vary considerably across Liberia. For example, there are wide disparities between the rights of male and female youth, as well as between the rights of youth who come from the original landholding families and those from land-poor households. Perhaps the most marginalized of youth are "stranger" youth who are not considered full community members. The extent to which Liberian youth face challenges in accessing land varies by area, though the scope of the problem is difficult to assess given the lack of available evidence.

The findings in this paper are presented using a Gender-Responsive Land Rights Framework [3], which is a conceptual tool that allows us to analyze and present nuanced information on the various dimensions of land rights, including how land rights are granted and realized by various actors in various contexts as well as how they are mediated by formal and informal institutions. The framework is used here to examine the rights of female and male youth in the context of rights to customary land in rural Liberia.

The findings highlight that youth face significant but different barriers to long-term access to land with female and "stranger" youth facing considerably more barriers to accessing land than male local youth. While the Land Rights Act has the potential to improve land access for youth and other marginalized groups, youth lack knowledge about land policies and traditional elders and elites still control land and resources in rural communities. Furthermore, Liberian youth are limited in their participation on land-use decisions and the crops they grow, and have limited access to credit and extension services, with female youth being less able to participate in land-use decisions than male youth. Youth's land tenure insecurity impacts their ability and desire to practice agriculture.

This paper is organized as follows: The next section reviews the data collection methods and explains the gender-responsive youth land rights conceptual framework used to present the findings. This is followed by a presentation of the findings using our conceptual framework. The next section discusses the findings and proposes recommendations to promote youth land rights in Liberia based on our findings.

---

[2]　"Strangers" are residents who are not in the direct landowning lineages of communities in Liberia. Usually they have various degrees of land-use rights but not ownership rights. Some of them, especially youth, are born in the communities, but because their parents are not in the landowning lineage they are often referred as strangers and have less secure land access and land rights.

[3]　The framework has gone through several iterations with contributions made by different stakeholders in the land sector. A version of the framework was used by Doss and Meinzen-Dick (2018), who cite Place et al. (1994) as originating the approach. An version of the framework was also used by Landesa to examine women's land rights in Myanmar, see Louis, Eshbach, Roberts and Htee (2018) "An assessment of land tenure regimes and women's land rights in two regions of Myanmar", Paper Presented at the World Bank Conference on Land and Poverty. Washington, D.C. March 2018.

## 2. Materials and Methods

The research discussed in this paper come from three separate research activities: (1) A review of evidence on youth and land access in Liberia; (2) a qualitative assessment on youth land rights conducted in August 2018 to inform the design of LRSD project for Landesa and its partners' community level interventions; and (3) a quantitative baseline survey of program beneficiaries conducted in October 2018 as part of an assessment of the LRSD program's outcomes on knowledge of community members' land right and attitudes toward women and youth land rights.

### 2.1. Literature Review

The literature review focused the available but limited evidence on youth access to land and the related obstacles they face in supporting their livelihoods in rural Liberia. The literature identifies several different obstacles that youth face in accessing land. These are discussed within the conceptual framework along with the findings from the primary research.

### 2.2. Formative Qualitative Research

The objective of the formative qualitative research was to understand youth and land issues in rural Liberia in order to inform the design of Landesa's Land Rights for Sustainable Development (LRSD) Project. The research focused on four major thematic areas: (1) youth livelihood activities, (2) youth access to land, (3) land-related disputes, and (4) youth land governance and community relationships. The research was conducted by Landesa in collaboration DEN-L, one of Landesa's civil society partners in Liberia.

Sixteen communities in Lofa and Bong County were included in the study and were purposively selected to represent the variety of land issues, cultural values, ethnic dynamics, and land tenure systems in rural Liberia. In addition, communities with proximity to concession activities as well as communities near urban areas were included, in order to better understand these dynamics.

A total of 16 focus group discussions (FGDs) and 46 key informant interviews (KIIs) were conducted. The FGDs included community members with a focus on youth, while KIIs included youth, government officials, customary authorities, and other leaders. Each FGD included nine to twelve community members and a total of 192 community members participated in the FGDs in both counties; 70% of the participants were "youth" (15–35 years old) and 44% were female. By occupation, 70% of FGD participants were farmers and 20% were teachers. FGD participants were recruited by asking elder and youth leaders to identify youth, using the community definition of youth. Some participants over 35 years self-identified as youth or participated because of interest in youth issues.

KIIs were conducted with national and regional government officials, land experts, and civil society organization leaders (CSO). In addition, at least two KIIs were conducted in each community, with traditional leaders, elders, CSO leaders, youth leaders, and other key land stakeholders. About 10% of KIIs were with youth stakeholders. By primary occupation, KII participants mirrored the FGD (majority famers, although 15% said CSO leader was their primary occupation). CSOs and opinion leaders were selected to balance information in the assessment, as many of them have no direct link or interest in community lands. KII participants were selected according to DEN-L's knowledge of the stakeholders in the communities. The KIIs generally discussed the same topics as the FGDs, but were geared to KII participants' unique knowledge, viewpoints, and unique roles/responsibilities in youth and land issues.

### 2.3. Baseline Survey

Landesa conducted a baseline survey with LRSD project beneficiaries located in seven communities in Bong, Lofa, and Rivercess Counties in October 2018 just before implementation of community-level activities through its CSO partners. These communities were targeted for land rights awareness activities implemented by Landesa's partner CSO organizations, DEN-L and FCI. The survey sought

to establish a baseline on beneficiaries' knowledge of and attitudes toward community membership; land rights, including the land rights of women and youth; land laws and institutions; land governance; and Alternative Dispute Resolution (ADR).

The research sample was exhaustive and targeted all beneficiaries selected to participate in the trainings. In some cases, community members who were not selected to participate took part in training; these additional participants were not included in the baseline survey. The beneficiaries of the training programs were selected by DEN-L and FCI staff through an intensive community consultation process in each of the seven program communities, which took into account community members' ability and willingness to participate in the trainings. The participants represented a broad spectrum of community members, ranging from traditional elders, leaders (including religious, women, and youth leaders), teachers, as well as a diverse spectrum of community members at large (including women and youth). Table 1, below, illustrates the age and gender of the participants/survey respondents in the seven program communities.

**Table 1.** Age and Gender of Baseline Respondents, Percentage in Program Communities.

| Community | No. of Respondents | Male | Female | Adult | Youth * |
|---|---|---|---|---|---|
| **Lofa County** | 85 | 62% | 38% | 49% | 51% |
| Pasama | 36 | 47% | 53% | 42% | 58% |
| Bardezu | 24 | 29% | 71% | 79% | 21% |
| Gbonyea | 25 | 32% | 68% | 32% | 68% |
| **Bong County** | 73 | 41% | 59% | 47% | 53% |
| Gbarnga Siaquelleh | 36 | 50% | 50% | 44% | 56% |
| Shankpowai | 37 | 67% | 33% | 50% | 50% |
| **Rivercess County** | 50 | 30% | 70% | 82% | 18% |
| Neezuin | 25 | 32% | 68% | 80% | 20% |
| Little Liberia | 25 | 28% | 72% | 84% | 16% |
| Total | 208 | 43% | 57% | 57% | 43% |

* Youth are considered those aged 15–34, which is based on the definition used by the Government of Liberia and the African Union.

## 2.4. Women's Land Rights Conceptual Framework

The data from the research are presented using a Gender-Responsive Land Rights conceptual framework modified to fit the context of property rights of youth and women in rural Liberia. The Framework defines property rights as multi-dimensional, and states they should be effective, inclusive, and Gender Equitable. Effectiveness is defined as (1) the bundle of rights enjoyed by an individual or group is clearly defined in all its dimensions for all members of a community, including those who may have new rights to that land (Completeness); (2) the duration of the bundle of rights are clearly defined and enforced (Duration); and (3) the bundle of rights are clearly enforced in the face of contestation (Robustness). In addition, (4) are all these aspects of rights Inclusive and Gender Equitable?

The framework takes into consideration the institutions through which rights granted by customary and statutory law are mediated. The institutions are:

- Statutory and customary land-related laws, policies, regulations, conventions, and agreements that embody the rights determined and enforced by governments;
- Formal and informal institutions and actors who influence, decide, manage, or enforce land-related rights;
- Social norms that shape attitudes and beliefs on who should have land, for what purpose and through which means; and
- Individuals and communities whose land-related rights are protected, strengthened, limited or negated by the system.

For this analysis, the framework is operationalized in the form of questions that seek to assess (1) how rights are defined in the law, (2) the level of awareness of rights and attitudes toward rights for female and male youth, and (3) the extent to which different groups of youth are able to realize their rights in practice within their socio-political contexts. The main dimensions that are addressed in this analysis are completeness, robustness, gender equality and inclusiveness. The dimension of duration is addressed in the completeness dimension for this study and is therefore not used. The dimensions, their definitions, and the questions used to operationalize the framework are presented in Table 2 below:

**Table 2.** Operationalizing a gender-responsive land rights framework.

| Land Rights Dimension | Definition | Questions Used to Analyze Findings |
|---|---|---|
| Completeness | Whether the bundle of rights enjoyed by an individual or group is clearly defined and enforced in all its dimensions for all members of a community, including those who may have new rights to that land | (1) Are the rights for female and male youth clearly defined under the LRA and related laws that govern land rights? (2) Do female and male youth and others have knowledge of youth land rights, and (3) Are female and male youth able to realize their rights? |
| Robustness | Whether rights can be enforced in the face of contestation | (1) Does the LRA clearly define enforcement of rights for female and male youth in the face of contestation, (2) Do female and male youth know how to enforce their rights in the face of contestation, and (3) Are female and male youth able to enforce their rights in the face of contestation. |
| Gender Equitable | Whether the land rights system give all the ability to participate in decisions about the land | (1) Does the LRA clearly define land rights/land governance as inclusive and gender-equitable, (2) Do youth understand their rights as they relate to decision-making on land governance? (3) Are youth able to participate in land decisions at the community level? |
| Inclusive | | |

## 3. Presenting Findings Using the Gender-Responsive Land Rights Framework

Using the framework, the quantitative baseline survey data are presented using descriptive statistics and are integrated into discussion of the qualitative findings and evidence from the literature reviews where relevant.

### 3.1. Is the Extent of the Bundle of Rights Clearly Defined and Enforced? (Completeness)

3.1.1. Are the Rights for Female and Male Youth Clearly Defined under the LRA and Related Laws That Govern Land Rights?

The LRA is a historic, comprehensive land law that provides new broad-based legal recognition of clearly defined bundle of rights to land for customary communities and individuals. Rights to access, control, and own land—collectively and individually—apply to female and male youth through robust equal protection provisions and inclusive community membership provisions yoked to recognized land rights.

The LRA incorporates fundamental principles of Liberia's Constitution and Land Rights Policy applicable to youth and their rights to land. Specifically, equal protection guarantees apply to all Liberians regardless of age and sex (art. 2). A core purpose of the LRA is to recognize and "ensure equal access and equal protection with respect to land ownership, use and management" (art. 3). This extends to giving equal legal protection to Customary Land ownership and Private Land ownership. Furthermore, the Customary Land rights bundle is clearly defined to include the right to exclude others, to possess and use land, to manage and improve land, and to transfer (art. 33). In doing so, the LRA recognizes and clarifies new rights to land for customary communities, including their members who are female and male youth.

Community membership is defined by the LRA in a way that includes all youth that meet the broad definition and requirements—regardless of age and gender. [4] Community membership broadly confers collective and individual rights to land. New generations born into the community also enjoy community membership and land rights (art. 34(5)). All community members meeting the LRA's definitions are part of the community and as such enjoy equal rights to use and manage their customary land, regardless of age or gender (art. 34(3)). All communities, acting collectively, control customary land through the power to make decisions about transfers of customary land to the government or non-community members (art. 36). Within a community's customary land, all community members, regardless of gender, have a right to exclusively possess a parcel of land for a residence (art. 39(3)). They also have the right to access land for agriculture (art. 40) and enjoy ownership and management of protected areas (art. 42(3)).

The LRA also recognizes private land ownership rights, held individual or jointly, enjoyed by all citizens, which includes female and male youth. Female and male youth also enjoy inheritance rights as sons and daughters including to parents in both statutory marriages under the Decedents Estates Law (§§ 3.2 & 3.4) and customary marriages under Equal Rights of Customary Marriage Law (§ 3.2). Finally, youth in the age range of 15 to 18 enjoy inheritance rights under the Children's Law (2011) (s. 17.1).

Despite the robust and inclusive provisions recognizing rights to (and rights within) customary land in the LRA and land-related framework, there are gaps that impact youth. For example, where patrilocal marriage is common combined with an increasing trend in marriage informality in Liberia [7], cohabiting youth, particularly female youth, moving into their partner's community, and residing there for under seven years, will either have less clear rights to customary land or no rights. In the first category, to be deemed a community member, the key to enjoying rights in customary land requires meeting the nebulous legal hurdle of qualifying as a presumptive spouse. [5] In the second category, youth from outside the community who are cohabiting with a community member but who do not meet the standard of presumptive spouse will not be considered community members, and as such will not enjoy rights to customary land under the LRA. Youth in these categories are land tenure insecure under the law.

3.1.2. Do Youth Have Knowledge of Their Bundle of Rights (i.e., Have They Been Clearly Defined for the Youth)? Are Youth Knowledgeable about Who Can Gain New Rights to Land and under What Circumstances? Is This the Case for All Youth?

Findings from the formative research highlights that youth in both Bong and Lofa counties know little about their land rights granted through statutory laws, including the Land Rights Act and the Community Rights Law. In general, youth are more knowledgeable about customary norms and laws regarding land rights. These are generally understood by youth as follows:

1. Males are the heads of households.
2. Elder males and landlords (male) have the most rights and privileges to land.
3. Land matters are for males, not females.
4. Elder males are the arbiters of land access and governance and need to be approached in the traditional way for land.
5. Youth must approach elders for land.
6. Youth can only access land within the community if he is a community member (i.e., his father is from the community).

---

[4] Under the LRA, a youth may legally qualify as a community member if he or she is a Liberian citizen and who fits in any one of the follow categories: (i) born in the community, (ii) parent(s) was born in the community, (iii) community resident of 7 years, or (iv) spouse of a resident community member (inclusive of statutory, customary, and presumptive spouses) (art. 2).
[5] "Persons who live together as husband and wife and hold themselves out as such are presumed to be married" (Civil Procedure Law, § 25.3(3)).

7. "Strangers" (i.e., parents were not born in the community) have more restricted access to land.

Some had heard about a few provisions of the Land Rights Act through radio programs. A few youth understood the difference between tribal certificates and deeds and recognized that those with deeds had stronger rights than those with tribal certificates. [6] Often, this understanding was the result of conflicts in which youth were involved, which resulted from people with deeds encroaching on land that had tribal certificates.

These findings were largely confirmed through the quantitative baseline survey. In the survey, respondents were asked if they had heard about the Land Rights Act, Marriage Law, the Constitution and the Land Authority. Responses indicate that only 55 percent of youth had even heard about the Land Rights Act, let alone understood any of its provisions. Almost twice as many males were aware of the laws, and more adults reported knowledge of the laws than youth, despite the higher education levels of youth. Figure 1 below highlights respondents' awareness about land laws and institutions by gender and age.

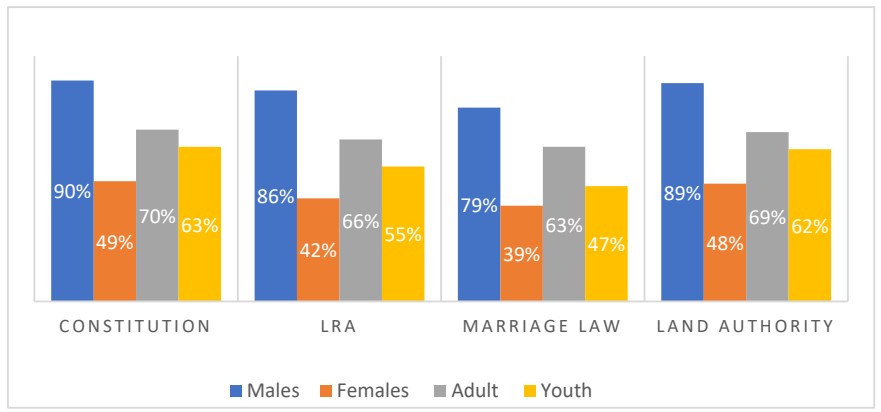

**Figure 1.** Community members' awareness of of land laws/institutions, by gender/age.

Knowledge Gaps on land rights emerged between women and men: women often answered they "didn't know" in response to questions on land rights; youth also often lacked knowledge on land rights (many of these were female youth), as exhibited in the Figure 2 below.

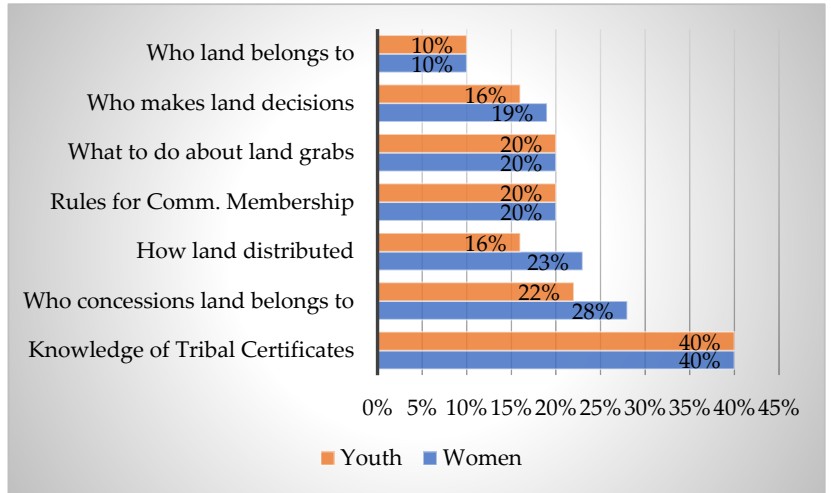

**Figure 2.** Youth and women who "Don't Know" about land rights under the Land Rights Act.

---

[6] Tribal Certificates are an initial step for registering land but until it is transformed into a deed, the land rights are only usufruct (use rights).

Views on women and youth land rights did not vary much by age, but they did by gender. Male respondents believed that youth have more land rights than did female respondents; for example, over 90% of men agreed to the statement in the survey questionnaire that "Youth who are community members have the right to make decisions about land" while only 58% of women agreed.

Interestingly, male participants also thought that women had stronger land rights than did female participants: almost 90% of men agreed to the statements that "Every woman has the right to participate in making decisions about land" and that "Daughters can inherit family land", while just over half of women agreed with these statements. In fact, men reported more positive attitudes and beliefs about the rights women should have than women did. Interestingly, nearly all of the respondents who said their communities made land decisions collectively reported that women had the right to participate in making decisions about land in their community.

In the formative research, a majority of youth understood that community membership is the prerequisite to accessing community land for farming or house plots, as well as for participation in land governance bodies. Youth understanding of community membership reflected current customary understandings that privileged those whose parents, or at the minimum whose fathers, are from the community. This understanding reflects customary norms and is different from the provisions of the LRA, which states that all (Liberian citizens) who have lived in a community for seven years, regardless of whether they were born outside the community, are community members. This provision puts "strangers" and their children, spouses (born in other communities), and wives in de-facto unions (without dowry) and polygamous marriages on equal footing with those who are traditionally considered community members.

The baseline survey results similarly highlight the disconnect between the legal definition of community membership in the LRA and the current norms around who is considered a community member. If one's parents were not considered official community members (i.e., the parents were "strangers") only 26% of the respondents said one would then be considered a community member, even if they were born in the community. Furthermore, only about half of respondents thought that living in the community for over seven years would make one a community member. In terms of who makes the rules for community membership, 60% of respondents said they are made by the traditional authorities; only 18% said they are made by the community as a whole. Figure 3, below, illustrates the beliefs around the various criteria for community membership. In the baseline survey, a high percentage of males (90%) reported that any resident, including women and youth, could plant life trees or cash crops, while fewer women (approx. 65%) felt this way. Similarly, more than half of males and adults felt that individuals of any age or gender could own land, but women and youth respondents were less sure of ownership rights, especially for themselves. This discrepancy between what men report and the actual practices on the ground is to be noted.

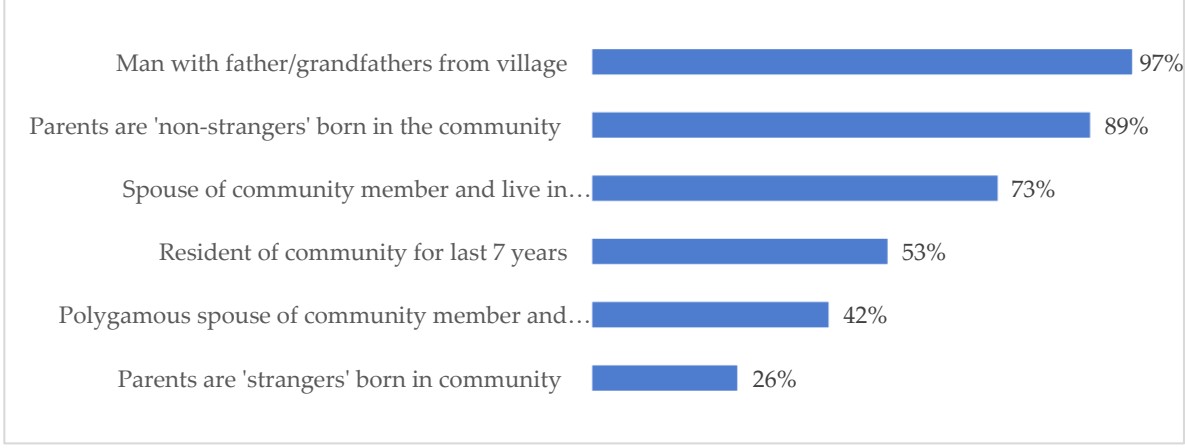

**Figure 3.** Community Members' beliefs on the criteria for community membership.

3.1.3. Are Youth Able to Enjoy the Full Extent of Their Bundle of Rights? Are Youth Able to Gain New Rights to Land If They Meet the Criteria for Eligibility? Is This the Case of All Youth?

For youth, family inheritance or allocation by customary authorities appear to be the most common means of accessing land for farming. In many clans, youth cannot inherit land or be allocated their own land to farm until they are considered "adults." Instead, they must work for their families and communities until this time, especially youth who are not born into families of the landowning lineages. Studies suggest that these practices are rooted in subsistence farming's reliance on surplus youth labor, especially in the northwest of Liberia, and the gerontocratic nature of many rural communities [8–10]. In addition, longer life expectancies of elders, a youth population bulge, and higher scarcity of land in some areas, has meant further delays and/or smaller land inheritances inheritances/allocations for youth. Delayed land rights prevent youth from planting "life trees," which are commercial tree crops like rubber or oil palm, through which one can assert a permanent claim to the land. These can only be planted on inherited land [11]. Smaller inheritances can mean parcels are too small for youth to earn a living. Increasingly, encroachment by outsiders on traditionally held lands has reduced the availability of land youth can receive, which is especially problematic in areas where valuable land is less plentiful [11].

The Eleven Clan Study found that in communities where land rights are vested in local authorities, the town or quarter chief would allocate newly married youth land for seasonal cropping [1]. These rights are typically for one or two agricultural seasons, with limitations on what crops can be grown. In communities where land rights are held by families, inheritance is more common. In some areas, informal rental markets provide an alternative for youth [1]. However, leases are often short-term with high rental fees, and cash cropping is not allowed.

Migrant youth faced extremely exploitive arrangements by customary authorities prior to the war, which fed youth into the militias [12] and the concern is that customary leadership might continue to exploit migrant youth who are considered strangers, particularly with regard to land access [12].

Some studies found youth were dissatisfied less in terms of access to land and more in terms of their lack of agency over the land (e.g., not being able to plant life trees). Male youth are increasingly challenging customary governance institutions by rejecting traditional practices, decisions made by customary authorities, and directives to contribute to town labor. Studies find that youth are increasingly showing interest in individual, documented land rights [9,13,14].

The primary avenue for youth access to land in Bong and Lofa counties is through their parents and traditional leaders. Most youth can typically access land only for short-term use (which allows farming of vegetables/grains) while long-term access to land for more profitable commercial crops or more permanent life trees is typically only granted to male youth from landholding families.

Male youth from landholding lineages have more secure access to land. They generally access land through their parents or family elders for farming or house plots. Even those who come from landholding families must wait to inherit land to enjoy their full spectrum of rights such as long-term access to plant life-trees. Male youth with older brothers may not inherit rights to land unless they are willing to work the land with them and are in good standing with their brothers. The traditional process of gaining access is for elders to recite the land history to youth which teaches the youth about their rights as well as the extent of their land boundaries and the nature of land-use. Male youth who are children of parents in polygamous marriages, may be denied rights even if they are from landholding lineages or if their parents own land, if they are not children of the formal wife.

Youth who are not from land-holding lineages may gain access by approaching elders in their community and sometimes from neighboring communities. Sometimes youth pay a nominal fee for use of the land or are expected to give small gifts to the elders who allocate land to them. Rights granted to these youth are largely for seasonal or short-term use for farming vegetables or grains and cannot be used for trees or cash crops because that would establish long-term rights to land. These youth may never gain long-term rights to land, unless they have the money to buy land.

Stranger youth face even more hurdles in accessing land. While they may be legally recognized as community members and hence have rights to customary land under the LRA, traditional norms tend to exclude these groups from accessing land. Depending on their situation they may be able to access land by finding a local elder to vouch for them ("stranger father") or leasing land. But their rights are less secure. While use rights to customary land are inherited, stranger youth mentioned that they were denied access to land even though their parents were allowed to use land. For example, some explained that they were told that land was not their parents in the first place, even if the parents had planted cash crops.

Some youth reported that those who lease land or are allocated land for short-term use have faced arbitrary evictions. Some youth also reported that they are not allocated land in time for planting or are given land that is not fertile. Sometimes, they are limited to how they can use land. For example, some youth mentioned that they are not allowed to plant certain crops or fish on the land.

Youth reported that land scarcity was one main reason that impacted their access. Increasing population density has increased demand for fertile land and house spots and made it less available for youth (mentioned in two communities in Lofa County, and two in Bong County). When land is scarce, stranger youth are more at a disadvantage. For example, stranger youth in Baila in Bong county, reported that they paid high rents for house plots because of land scarcity. Some youth mentioned that landlords sell-off land to outsiders, therefore there is not enough land for the youth. In some cases, most of the available land is taken for life trees, leaving little land for youth to cultivate vegetables.

### 3.1.4. Are Women Able to Enjoy the Full Extent of Their Bundle of Rights? Do Women Have Knowledge about Who Can Gain New Rights to Land and under What Circumstances? Is This the Case for All Women?

"Men have the right to land; women get permission from men because they (men) are the head and leaders of the family"—Male Focus Group Participant.

"Land business is a traditional thing so women cannot own land here. Besides, they do not even have the strength to fight or defend the land when there is conflict"—Male Focus Group Participant.

Most research respondents report that women's rights to land are derivative of men's rights. Single women gain access to land through their fathers, brothers, uncles, and other male relatives. Customary norms do not treat young single women as full members of the community because they are expected to eventually marry into a family from another community as is the practice in patrilocal systems. Married women gain rights through their husbands, and widows do not automatically acquire rights to their husband's land, rather they gain access through their deceased husband's brothers or other male relatives who control the land.

In Bong and Lofa counties, female youth are even more disadvantaged than male youth in access to land for agriculture or house spots. In one FGD, single female youth explained that they are rarely able to access land directly, and need to "beg" their brothers, fathers, uncles for land. Landholding families fear that control of their community land will pass to outsiders and therefore do not allow their daughters to inherit land.

Even when women do access land, their marital status determines the extent and duration of the bundle of rights they can enjoy. Single young women may access land for short-term access. For example, in Lofa County, Sucrumu village, single women from land holding families are actually able to access farming land easier than married women; however, these short-term rights are usually terminated when women get married. Long-term access for married women is mostly through male relatives. When women community members marry strangers, they may receive only short-term rights to land within the community, but their children may be recognized for long-term rights. The situation of women in de-facto (marriage without dowry) and polygamous marriages is not clear from this research.

In the survey, similar to the findings in the literature and the qualitative research, most respondents said women typically go through a husband or male relative to access farmland (only 25% said a

woman could get land without such permission, and only 37% said female youth could independently access community farmland). De-facto unions appear to further disadvantage women's land access and use over more formal arrangements: while over 80% of respondents said women in customary and statutory marriages have the same rights to use customary land as their husbands, only 60% said women in de facto unions have the same rights. Respondents felt youth had rights to community land, but it seemed marriage may be a pre-requisite: while few respondents felt "land was only for adults," (15%), only 47% of respondents said unmarried youth are co-owners of community land, while nearly 60% said married youth are co-owners.

In the survey, almost all participants responded that if a woman's parents were not strangers (parents who were born in the community), she was also considered a community member. On the other hand, only 72% said being married to a community member made one a community member. Considering women often relocate from their natal area to their husband's community, this may indicate that women are not considered "members" of the communities they live in after marriage. Women in polygamous unions appeared to be further disadvantaged; only 42% of respondents said a woman in a polygamous union (even to a community member husband) would be considered a member. Interestingly, in these more ambiguous cases of community membership (e.g., stranger parents, polygamous unions), women and youth respondents were even less certain that one would qualify for membership than adults and men were. As mentioned earlier, land access is contingent upon community membership, therefore ambiguity about who is a community member impacts their rights to land.

In the formative research, some youth shared examples of women accessing land directly either because they are from landholding families and they get land from their fathers, or they have the funds to rent or buy land. For example, in Baila Village in Bong County, one woman who is a nurse bought her own land and has planted life trees such as rubber. In Kpaai, some families allow their girls to own land; the context under which this happens is unclear from the research. When women gain rights by renting land, access is usually clearly defined as short-term. Access for house spots are given as long-term if women have managed to access them (women need to approach traditional elders through male relatives for house spots).

However, most youth in the formative research highlight that traditional norms do not allow women to be involved in land matters. These youth believe that it is hard to change customary norms that limit women's access to land. In some cases, even if women are given short-term access to farmland, their rights are not respected, and others encroach on their lands. This was mentioned in an FGD in Yeala village in Lofa County. Furthermore, young women are sometimes not able to use the land they are given because they lack access to labor and other inputs. When the land they use is in the bush or far from the village settlements, women cannot go to the boundaries of the land (the bush) on their own because of safety reasons.

House Spots: Although community farmland is typically thought of as collectively owned, plots designated for homes (house spots) seemed to be considered individually owned; nearly all survey respondents (from both genders and age groups) said that community members were able to claim a house spot "forever," even women and youth—but women needed the permission and aid of their father or husband to claim a spot. However, almost all respondents said that women could inherit a plot, pass it on to their children, and that a widowed woman can stay on the spot. In terms of youth, nearly 70% said youth (male and female) can inherit family land, but in contrast, only 37% said that female youth can independently own land.

*3.2. Are the Bundle of Rights Clearly Enforced in the Face of Contestation (Robustness)?*

3.2.1. Does the LRA Clearly Define Enforcement of Rights in the Face of Contestation

The LRA contains high-level provisions that define the enforcement of land rights in the face of contestation. It obligates the Government of Liberia to protect and enforce all land rights and interests,

which includes female and male youth rights to land (art. 10). An explicit objective of the LRA is to guarantee that all communities, families, and individual "enjoy secure land rights free of fear that their land will be taken from them, except in accordance with due process of law" (art. 3). Critically, Customary Land's existence and ownership is enforceable by operation of the LRA (art 32(2)), [7] and not affected because of lack of title or registration (art. 11(3)) or to lack or delay in completing the confirmatory survey of customary land boundaries across the country (art. 37(4)). These provisions give more legal force to the claims of individuals and communities, including the youth members, in the face of conflicting interest by government, companies, and private parties. Furthermore, LLA decisions (that lack objections and exceptions) can enjoy judicial enforcement, subsequent to filing a petition, and all LLA decisions are subject to judicial review through the Circuit Courts and appealable to the Supreme Court of Liberia (art. 37). The LRA also requires the government to provide adequate resources to implement the LRA's legal provisions (art. 36(13)).

The LRA also has high-level provisions to define enforcement of rights in the face of contestation at the community level. A community member's absolute right to his or her residential area is automatically transferred by operation of the LRA and requires the CLDMC to formalize these rights by issuing a deed, the absence of which does not defeat a community member's ownership rights (art. 49). The LRA prohibits a community from depriving a community member of his or her residential area. Community restrictions on the exercise of land rights by a community member, including youth members, are invalid if they violate the LRA or Constitution (art. 34(4)). [8] The community and its community members are jointly and severally responsible to protect the rights of private land owners located within customary land; this includes private land owners who are youth community members who have a right to own a residential parcel. Private landowners, including those who are non-community members, are also responsible to abide by community-adopted rules (art. 46(3)).

Finally, the LRA is a comprehensive land law that envisions more detailed enforcement provisions to come under implementing regulations. The LLA holds the authority to promulgate regulations necessary to effectively implement the LRA (art. 71). The LRA specifically requires the LLA to develop regulations to solve all customary land disputes between communities through customary law and alternative dispute resolution (ADR) mechanisms (art. 37(8)), including those available through customary ADR bodies. [9] With the LRA recently passed, the LLA is in the process of developing all necessary regulations. These regulations will determine how well and clearly female and male youth land rights will be enforced in the face of contestation. More clearly defined enforcement will hinge on how robustly LLA promulgated regulations respond to youth and gender considerations.

### 3.2.2. Do Youth Have Knowledge of Their Rights in the Face of Contestation or Changed Circumstances? Is This the Case for All Youth?

Youth were generally aware that they can approach elders or the courts to deal with land conflicts but were largely in favor of settling disputes outside of court because they reported that using courts are expensive and disputes may take many years to resolve. Some youth in the FGDs reported that disputes in their communities involving tribal certificates are settled by elders, while those involving deeds are generally settled in the courts.

Youth's faith in customary leaders to resolve conflicts was mixed. Some youth trusted the traditional authorities and felt that elders were open to their concerns and listened. However, some said that elders are biased toward landowning and wealthy families and disputes were usually settled in their favor. Some youth say that they do not have a voice with elders.

---

[7] For example, as opposed to being enforceable only after the completion of an administrative process.
[8] Although land can be taken if the community provides the community member with comparable land.
[9] Under the LRA, alternative dispute resolution mechanism is defined as any process used to resolve disputes outside of court, and alternative dispute resolution body is defined as any entity, including a customary body, whose purpose is to resolve disputes outside of court (art. 2).

Many suggested that the best way to avoid land conflicts is by surveying and demarcation of land boundaries. For example, in two communities in Bong County, youth said that after land was surveyed conflicts came to an end. They also thought it was beneficial when fathers and elders show youth the land boundaries to help prevent encroachment and disputes. To better equip them to claim their rights, some youth expressed a desire to understand the provisions of the LRA; they also said they would like NGOs or the government to be involved.

While in some communities youth and elders have good relationships, many youth in the FGDs said that elders do not understand youth's needs and that elders need to be sensitized to the problems of youth and their need for land and also help build relationships between youth and elders. In a few FGDs, some youth thought that the government and NGOs should bring youth and elders together to train them and promote a dialogue, so that youth can better access land within the community. Some youth expressed the need to be more involved in land matters in order to increase their access to land.

In the survey, when asked what they can do about grabs of community land, the overwhelming majority of male survey respondents reported they could sue individuals (90%) and companies (83%), and half even said they could sue the government; for females, the percentages were much lower. However, 66% of respondents said the land that big companies are currently using for concessions belongs to the communities.

### 3.2.3. Are Youth Able to Enforce Their Rights in the Face of Contestation or Changed Circumstances? Is This the Case for All Youth?

There is little information in the literature about youth participation in customary dispute resolution mechanisms; this could be an area for further study. In general, there is a deep distrust of customary courts and their capacity to decide land issues in a fair manner [12]. The Eleven Clan Study found dissatisfaction with the customary system appears to be particularly prevalent among youth. In many clans, local authorities are viewed as biased, especially by the youth, who question the legitimacy of these authorities [1]. Youth did not typically hold significant positions of power in the process [1].

From the formative research we learned that contestation and conflict over land in Bong and Lofa Counties are characterized by: (1) Encroachment of boundaries between landowners and between communities, often exacerbated by lack of fertile land and unclear boundaries, which can inadvertently lead to encroachment as youth clear land in the bush for agriculture. Some research participants complained that tribal certificates do not protect their land from encroachment, (especially from those with deeds). They report that only deeds protect land and only those with deeded land can sell their land. However, most residents said they did not have deeds. In FGDs, many youth explained this was due to the difficulty and expense because land must be surveyed, and the cost is prohibitive; (2) intra-family disputes over land, including multiple claimants for family or tribal land and contestation between "legitimate" and "illegitimate" children for rights to their father's land; (3) disputes caused by people selling the same land to more than one buyer; and (4) disputes caused when people who have only short-term access to land to plant vegetables instead plant life trees to make a claim on the land; in some cases people even do this on land that is set aside for communal use; (5) ethnic conflicts were reported in two communities in Lofa county.

Most participants report that the most common way to solve land conflicts is to involve the elders who determine the rights based on their knowledge of the history of the land. In some cases, youth are also involved in settling community land disputes; this was mentioned in three villages in Lofa County and one in Bong County. There is a hierarchy of traditional elders that are involved in settling conflicts. First the local community elders and landlords are asked to arbitrate. If that is not fruitful, town and quarter chiefs are approached. Other institutions have also played a role in settling land conflicts. For example, in one community, the Liberia Refugee Repatriation and Resettlement Commission (LRRRC) was involved in settling land conflicts.

Both intra-family and community boundary disputes appeared to have a negative bearing on youth access to land in both counties. Some youth who participated in the formative research have experienced conflicts over boundaries fear violence and have no choice but to pursue other livelihoods or migrate to urban areas rather than fight for their rights. In two communities in Bong County, youth mentioned that conflict between neighboring towns or villages had even led to the loss of lives. Furthermore, youth say that they are the most affected by land conflicts because they do not gain rights to land being contested. In some cases, youth from families or communities who are in conflict cannot work together to use land.

3.2.4. Are Women Able to Enforce Their Rights in the Face of Contestation or Changed Circumstances? Is This the Case for All Women?

From the formative research, most youth reported that female youth are generally not involved in settling land conflicts. They explained that since men are mostly involved in clearing the bush, it is the men that encounter conflicts. However, in one village in Lofa county youth who participated in the formative research reported that elderly women leaders are allowed to help in dispute resolution. Some youth mentioned the need to organize themselves so that both male and female youth can be trained in conflict resolution.

*3.3. Does the Land Rights System Give All the Ability to Participate in Decisions about the Land (Inclusive & Gender Equitable)?*

3.3.1. Does the LRA Clearly Define Land Rights/Land Governance as Inclusive and Gender-Equitable?

The LRA formally devolves customary land governance to communities, including to their female and male youth members. Inclusive community membership provisions, combined with equal protection provisions, discussed above, ensure that the full community membership is included in community-level land governance bodies and holds customary land governance rights. The community membership, comprised of all community members acting collectively, is the highest decision-making body and is vested the authority to develop and manage customary land (art. 36). Female and male youth, as part of this collective, share this authority. Additionally, all Community Members, including female and male youth, are responsible for developing community by laws and a land-use plan and establishing Community Land Development and Management Committee (CLDMC) (art. 35).

The CLDMC is a community-level (executive) land body comprised of equal representation of youth, women, and men, who share decision-making authority (art. 36(6) & (7)). The CLDMC is accountable to the entire community membership, including its youth members (art. 36(4)). Even prior to the formal establishment of the CLDMC, youth and women along with elders, chiefs, and traditional leaders are responsible for developing and managing its customary land (art. 69).

While provisions in the LRA include female and male youth in community-level land bodies and grant them land governance rights, there are gaps. Whether female and male youth are able to exercise these governance rights, particularly when all community members are acting as a collectively, will depend on how well and inclusively LRA regulations operationalize these rights. Additionally, similarly to the issue raised above, cohabitating youth who move into their partner's community, and who reside there for under seven years, will either have less clear rights or no rights to participate in customary land governance.

3.3.2. Do Youth Understand Their Rights as They Relate to Decision-Making on Land Governance?

In general youth did not understand their rights to participate in decision-making on the land, but we did not learn anything specific to this particular aspect of youth rights. Youth however did indicate that they were unhappy to not be able to participate in land governance decisions. See following section.

### 3.3.3. Are Youth Able to Participate in Land Decisions at the Household and Community Level? Is This the Case of All Youth?

Youth participation in decision-making and land governance is variable. There are national, regional, and local youth organizations that advocate for land and resource rights on behalf of youth [1,13,15,16]. The Eleven Clan Study reported that in many clans, youth are well-respected and take part in land decision-making, the formulation of rules, and sometimes even in resolving disputes, though the level of involvement varied significantly [1]. However, in other clans, youth felt overlooked in decision-making about land issues, and were excluded from traditional governance structures. Some youth have expressed frustration about their lack of power in community land management processes [1,13,14] which has fueled a sense of exclusion and resentment [1,9].

Rural youth participation and control in local governance may have increased since the civil war. In areas that grant concessions, new governance structures have been established under the Community Rights Law for Forested Land [8] to negotiate with concessionaires. Even though some youth are represented (by requirement) on these governance structures, their participation may be weak [8,17]. In a recent study on women's participation in forest governance bodies conducted by Landesa in collaboration with the LLA and FCI, in four Community Forest sites in Bong, Grand Bassa and Margibi Counties, male and female youth seemed to be well represented on the forest governance bodies, especially in the Community Assembly. However, decision-making around dealings with companies or distribution of monetary benefits are often by elite members in the Community Forestry Management Body [18].

During the formative research, many research participants reported that male elders from landholding families have a firm grip on land related decision-making processes, while those that do not have land are excluded from governance decisions. In most cases, only select youth (usually youth leaders, or land-owing youth) are allowed to be part of land governance and that it is the elders' decision on whether youth are involved in land matters; youth cannot participate unless they are invited. Even if youth are involved, they felt that their voices do not carry as much weight, and that elders have final say on land governance decisions. They believe that elders fear of losing control of the land and therefore limit the scope for effective youth engagement in community land governance systems. Furthermore, they say that elders think that young people are not serious enough to be involved in governance. Most youth expressed a desire to be involved more fully in land governance decisions working in collaboration with the community elders. They felt that they would benefit from helping facilitate elders and youth to working together.

In many communities in Lofa, the relationship between youth and adults on land matters was mixed. In some communities, youth and adults were reportedly working together very closely when discussing or resolving key community land issues. However, this was questionable especially given that some youth were hesitant to discuss land issues in the absence of elders. For example, youth leaders in one town would only discuss land issues when a community elder could listen and respond to some of the questions. It was clear that some youth in this town could not discuss or engage in any meaningful land-related issues without prior approval from the elders. In another town, community elders viewed youth as a big problem and expressed that youth needed to be punished severely for being lazy and for concentrating on drugs and other community vices. A youth leader however strongly suggested that "youth were in charge" of the community and he praised the youth for being vigilant in the face of border threats posed by migrants from Guinea. In that same community, however, key informant interviews indicated that it was difficult for female youth to access land. In another community, youth-adult tension was very high and there was no indication of youth-adult partnerships or dialogue platforms on land matters. In many Lofa and Bong County towns, the existence of young quarter and towns chiefs depict encouraging signs regarding the future of youth engagement in community land governance processes in Liberia. The young quarter and town chiefs appeared to have a good working relationship with community elders and a good understanding of the key land issues affecting youth in their towns.

As illustrated in Figure 4 below, about half of the survey respondents reported that traditional authorities made decisions about land. Nonetheless, nearly a quarter said the members of the community made land governance decisions collectively—and the majority of these said that this collective decision-making was inclusive of all community members. None of the respondents reported that a statutory land management body made land decisions [10].

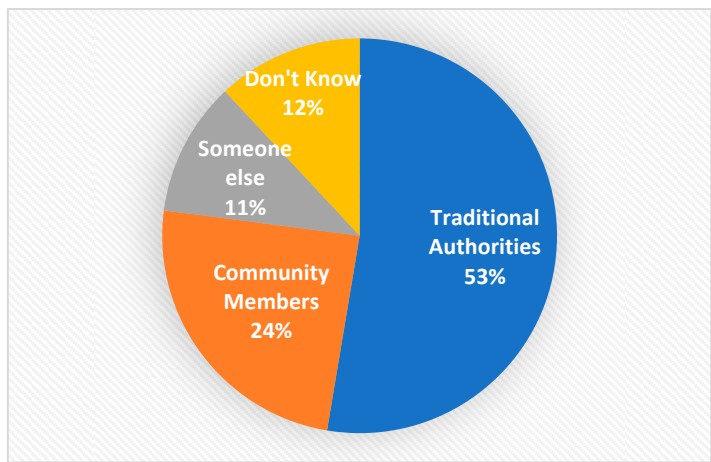

**Figure 4.** Community members' perception on who makes decisions about customary land in the community.

Because youth are often not landowners, and because they often lack intimate knowledge about land rights, community boundaries and the land governance systems, the research and literature found that they are generally excluded from land dispute resolution processes—including Alternative Dispute Resolution (ADR) systems, locally run systems set up by communities and/or CSOs to address unresolved disputes (typically after efforts by a traditional authority, or in lieu of them altogether). The formative research found that opportunities for community elders and youth to discuss and resolve land disputes are rare in both counties, which suggests that intergenerational tensions over land could increase in future, and force even more youth to abandon agriculture altogether.

3.3.4. Are Women Able to Participate in Land Decisions at the Household and Community Level? Is This the Case of All Women?

Female youth, in particular, are disempowered in terms of land governance, as the traditional view that land is the domain of men persists in rural Liberia. The Eleven Clan Study found that in some communities there are positions which have a strong influence over land that cannot be occupied by women. Dodd et al. [7] found that even where women held positions of authority, their power was often curtailed by traditional gender roles. A national report on youth engagement [19] found that young women are often the least engaged in their communities, hesitant to engage in politics, and overlooked by policymakers.

The formative research found that young women were excluded from land governance systems in deeply traditional communities, especially in parts of Lofa County, which could be a significant challenge for programs seeking to advance gender-equitable land rights for youth in these communities. When youth are involved in land governance, it is usually male youth that participate. There are some exceptions. For example, in one community in Bong County, youth in the FGD mentioned that young

---

[10]　These responses are for a question asked about 'big' land decisions,' i.e., regarding land over 50 acres; responses were nearly identical on a question about land under 50 acres—The only exception being that when it came to decisions about small parcels, about 5% said the family/clan made land decisions.

women participate in land governance but are expected to act in a deferential way when they speak. In another in community, women were involved in a town hall meeting for the construction of a clinic.

Youth—female youth in particular—are often unaware of their legal rights to land, including the Land Rights Act (LRA). Unsurprisingly, youth in both Bong and Lofa Counties were more knowledgeable about the exclusionary, male- and elder-dominated customary norms and laws regarding land rights. These customary land rights practices revolve around adult males as the arbiters of land access and governance with limited opportunities for independent youth access to land.

In the formative research the general opinion of research participants is that women are not expected to be involved in land matters according to traditional norms. Women's voices are represented either through their male relatives or the chairladies. However, in the FGDs, many youth acknowledged that women should be empowered to participate in decision-making about land governance.

## 4. Discussion

### 4.1. Impacts of Land Tenure Security on Youth Livelihoods

Land tenure insecurity impacts rural youth's ability to sustain a livelihood based on agriculture. Our findings highlight that most youth can typically access land only for short-term use which allows them to cultivate vegetables and grains, while long-term access to land for more profitable commercial crops or more permanent life trees is typically only granted to male youth from landholding families.

Agriculture is the primary income activity for youth among the formative research participants in Bong and Lofa counties. Youth were involved in commercial cash crop farming (e.g., palm, rubber) as well as vegetable farming ("gardening"). In Lofa, youth often worked in *kuus* and were occasionally hired as daily contract workers. Youth's agricultural work was divided along gender lines, with men doing more of the physically demanding tasks such as clearing brush. Men also seemed to control the more profitable cash crop farming while women pursued gardening. Besides farming, male youth were involved in driving motorcycles for hire, especially in Lofa, and youth sometimes had small informal businesses; in Bong, women are involved in petty trading.

Youth in Bong and Lofa counties have few livelihood options and the majority wished to deepen their engagement in agriculture to better support themselves and their families. However, a majority of youth felt that lack of access to land or short-term access impacted their ability to farm and support themselves and their families; many end up working as agricultural laborers.

Gender was an important factor in determining access to land and land-based livelihood opportunities for rural youth. Young women's access to land for farming and housing was even more constricted than male youth's access. Findings from the formative research highlight that single women who are given access to land are usually not allowed long-term use and their land rights are often violated and less respected. Even when young women were able to access land, they faced hurdles in organizing labor and capital to use the land.

While land access posed as an impediment to earning a living in agriculture for youth, lack of support to buy inputs also emerged as a challenge to youth's pursuit of agricultural livelihoods. Poor transportation and access to markets, low crop prices, and poor weather also were challenges. The most often-cited suggestion for improving youth participation in agriculture was to support access to agricultural inputs, along with agricultural training and supporting kuus.

These findings are consistent with the literature that highlights that limited access to land impacts youths' ability to practice land-based livelihoods. Agricultural employment is informal and vulnerable to downturns, so most youth complement it with other activities, such as mining and plantation employment, transportation services, government work, and small businesses. Most depend heavily on forest products for consumption and sale [1]. One study found that most youth migrate because of the inequalities in accessibility to land for farming [20]. However, with nearly one-third of youth unemployed (2008 census), urban economic opportunities are also limited. Many of the most marginalized youth are not suited to work other than in the agrarian sector [9]. The lack of

non-agrarian employment opportunities for youth, many of whom have had some forms of military experience from the war, is a source of social instability. Some engage in quasi-professional violence and illegal mining [21].

There is a widespread observation in the literature that Liberian youth are not interested in pursuing agriculture, for myriad reasons: they do not perceive that it offers a viable livelihood, the returns are too delayed as compared to daily work, and females may fear assault when farming far from home [11,22–24]. However, these perceptions are not backed by significant research, and some interventions and youth outreach officers report that youth are interested in agriculture, given the right conditions to earn a livelihood, a finding that is backed by our formative research findings.

## 4.2. Recommendation for Policy and Programming

The Land Rights Act presents a unique opportunity to recognize the rights of youth to access land and participate in land governance. However, closing the gap between statutory law that recognizes youth rights to land, and customary practices that are discriminatory toward youth, needs rural women and men to understand their rights under the LRA and buy in from the traditional gatekeepers who control access to land.

Our research in Bong, Lofa, and Rivercess counties provide useful policy and programming insights for promoting effective and gender-equitable youth land rights in rural Liberia. Landesa's ongoing LRSD interventions with CSO partners DEN-L and FCI are crafted to address some of the youth land access and governance challenges identified thorough our research, however, these are still at a relatively small scale, currently working directly in only 14 communities on the ground. To ensure impact at scale, policy and programming options might include:

- Land rights CSOs should co-develop and co-implement cross-sectoral youth livelihood and empowerment programs that seek to address not only youth land access and ownership challenges but also limited youth access to inputs, markets, poor transportation, low crop prices, and weather related challenges that discourage Liberian youth from pursuing a career in agriculture. The cross-sectoral youth and land interventions should also include nationwide radio and television programs showcasing successful youth farmers who can act as role models and drive home the idea of "agriculture as a business" and viable employment opportunity for Liberian youth.
- Supporting land rights education and awareness-raising to enable rural Liberian youth to better understand and defend their land rights as provided for in local customary and national statutory land rights frameworks. Community sensitization on the importance of youth land rights for sustainable rural youth livelihoods would be beneficial. The sensitization should especially target traditional gatekeepers such as customary leaders, landlords, elders, and local government officials, including town and quarter chiefs.
- Developing simplified language to popularize the provision of the Land Rights Act through online platforms such as WhatsApp and Facebook to increase the number of youth who are aware of the Land Rights Act and its provisions.
- Providing targeted and continuous youth land rights training at all levels of the traditional power structures and formal land administration systems to gradually break the deeply entrenched and oftentimes discriminatory land allocation and ownership systems that favor adult men and young men.
- Providing simplified land rights training and awareness raising Training of Trainers (ToTs) and community-based programs to grassroots-oriented youth associations, women's groups, and Community Based Organizations working with young farmers with particular attention to local organizations working with young women.
- Addressing ambiguities and questions around "who is a community member" by educating community elders, youth, and their representatives organizations about the community

membership qualification provisions under the Land Rights Act to ensure that the community and youth's understanding of community membership evolves toward a more inclusive membership criteria that protects the land rights of youth "strangers" and their children, and spouses and wives in de-facto unions and polygamous marriages.

- Promoting adult-youth dialogue on land matters in order to improve opportunities for youth to access land. This could be done by strengthening existing community platforms for youth-adult dialogue where they already exist (e.g., Yeala in Lofa County), as well as developing and supporting new youth-adult land rights dialogue forums where these do not exist. These dialogue forums could enhance youth engagement in broader community land governance systems and boost opportunities for youth access to land.

- Promoting youth-adult dialogue on land matters by organizing community-, county-, and national-level multi-stakeholder youth land rights workshops bringing together youth leaders, customary authorities, government officials, private companies, and development partners to educate stakeholders about the importance of youth land rights, youth related provisions in the LRA and innovative approaches to foster youth land ownership, and effective engagement of young men and young women in land related decision-making processes.

- Developing youth-oriented land rights advocacy initiatives based on thorough youth segmentation and considerations for most disadvantaged groups of youth including strangers, unmarried youth, females, and those in de-facto unions and polygamous marriages.

- Developing and promoting youth sensitive land rental market policies to better protect the land rights of youth from unscrupulous landlords and traditional leaders especially in Bong County where there is an informal, unregulated land rental market.

- Encouraging the Government of Liberia to develop community-level social security schemes that can incentivize and quicken intergenerational land transfers.

- Land rights CSOs should work with customary authorities and government officials to prepare a database of vacant or underutilized land that can be reallocated to prospective youth farmers especially those residing in and around major urban centers like Gbarnga and Voinjama. The database can be shared with youth associations to help identify landless youth who are interested in farming and connecting the youth to landowners or allocation authorities.

- Training youth leaders in land dispute resolution mechanisms including Alternative Land Dispute Resolution to build the capacity of youth farmers to recognize and defend their land rights when they are infringed upon by customary and statutory authorities. Land rights CSOs should work with the Liberia Land Authority to ensure the development of regulations that stipulate quotas for youth engagement in customary land dispute resolution mechanisms including Alternative Dispute Resolution (ADR) approaches. The regulations should define clear procedures for the protection of male and female youth land rights.

- Engage youth and their representative organizations in community boundary harmonization, mapping and surveying processes to ensure that youth have a good understanding of the historical evolution of land issues in Liberian communities where boundary disputes and intergenerational tensions over land are common.

- Civil society organizations should work with customary authorities and community leaders across the country to ensure the participation of male and female youth in the collective community development of bylaws, land-use plans and Community Land Development and Management Committees (CLDMCs). The community bylaws should indicate collectively agreed quotas for male and female youth participation in land-use planning processes and representation in CLDMCs.

- Building the capacity of grassroots-oriented rural youth organizations by educating them about land rights and linking them up with customary and statutory authorities that mediate rural youth access to land and opportunities for land-based livelihoods.

- In Bong and Lofa counties, communities where there is increasing land scarcity especially close to major urban centers, identifying and promoting profitable small-scale, intensive farming activities that target landless unemployed, and connecting the youth with land owners and land allocation authorities to promote group leasing, contract farming, and access to urban markets to improve youth access to land and quick income generating opportunities.

## 5. Conclusions

Our paper documented rural youth livelihood opportunities and land access and governance constraints for diverse groups of young men and young women using evidence from select communities in Bong, Lofa and Rivercess counties of Liberia. The paper highlighted that rural youth engage in a wide of range of livelihood activities of which agriculture is an important activity. For youth who want to engage in farming, it emerged that secure and long-term access to land is a major challenge especially for young women and youth who are considered strangers in the community. In all three counties where we conducted research, the land access challenges faced by youth are compounded by the existence of a customary land rights system dominated by adult males who often exercise a firm grip on land-related decision-making and land access opportunities. While most youth have a good understanding of the exclusionary customary land rights system, they do not seem to have a full understanding of their own land rights within the system and there are limited opportunities for youth to defend their land rights in non-confrontational ways. We noted how the lack of youth-adult forums to discuss land issues along with few opportunities for youth to participate in community land governance further constrain opportunities for youth access to land. In some communities, the continued exclusion of young men and young women in community land governance systems increases intergenerational tensions over land and further limits opportunities for youth to engage in farming.

Policy and programming options must address the individual and community level barriers that impede rural youth's access to land. To address individual-level youth barriers, land rights education, and awareness raising is critical. To improve youth access to land in deeply traditional settings such as Lofa County where land is communally owned, community sensitization targeting traditional leaders and landlords is fundamental. Youth land rights programs should be designed with particular attention given to gender, identity, and age dimensions that determine youth access to land. The success of these programs will largely depend on the extent to which they can help bridge the gap between statutory provisions and customary norms and practices that continue to discriminate against young men and young women in most parts of rural Liberia.

**Author Contributions:** Conceptualization, all authors; methodology, E.L., E.U.; T.M., M.-L.D., P.D. and T.H.; formal analysis, E.L., T.H., M.-L.D., T.M.; writing—original draft preparation, E.L., M.-L.D., T.M.; writing—review and editing, E.L., E.U.; visualization, T.H.; supervision, E.L.; project administration, E.L.; E.U.; funding acquisition, Landesa. All authors have read and agreed to the published version of the manuscript.

**Funding:** This research was funded by Kings Philanthropies through Landesa's Land Rights and Sustainable Development (LRSD) Project. The APC was funded by Landesa.

**Acknowledgments:** We would like to acknowledge the organizations and individuals that contributed to the research presented in this paper: The community members who participated in our research in Bong, Lofa, and Rivercess counties and the local and regional government officials and traditional leaders who gave us their time for Key Informant Interviews and supported our research in other ways. Our CSO partner DEN-L who conducted the formative research on youth and land. Special thanks to Peter Dolo for his leadership in the formative research and Dorothy Toomnan for conducting research in Bong County. DEN-L also facilitated our baseline survey data collection activities in their intervention communities. Our CSO partner FCI who facilitated our baseline survey data collection activities in their intervention communities. Research contractor Ecomsult, who conducted our Baseline Survey in Bong, Lofa and Rivercess counties. Special thanks to Jeremiah Collins and his team for doing an excellent job with data collection under challenging field conditions.

**Conflicts of Interest:** The authors declare no conflict of interest. The funders had no role in the design of the study; in the collection, analyses, or interpretation of data; in the writing of the manuscript, or in the decision to publish the results.

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
