# Peer review of "Using a Gender-Responsive Land Rights Framework to Assess Youth Land Rights in Rural Liberia"

_land, doi:10.3390/land9080247_

Round 1

Reviewer 1 Report

Dear Authors,

General comments

The manuscript (MS) deals comprehensively with the situation of young people and women, especially with regard to land rights, in the population of different counties in Liberia. Undoubtedly, in general, socio-cultural information related to land use in different African countries is often scarce and fragmented or, in any case, poorly understood. LANDESA's historical presence and commitment to the people of Liberia strongly supports the validity of this work.

The study consists of two phases, a qualitative and learning or information gathering phase and a subsequent quantitative phase based on survey-interviews with different stakeholders. The feedback between the two phases is very interesting as it allows for adjustment of the survey questions and through the survey responses adjusts the knowledge on the perception of the villagers on certain issues, especially those related to land tenure, gender and age.

Throughout the MS, detailed information is provided on Liberia's recent history and its impact on the specific problems raised. The MS is very well written and structured, perhaps the length of the text is excessive, which may detract from the strength of the message. Perhaps the literature review and framework could have been reduced and a more detailed analysis of the data (quantitative phase) could have been made and would certainly have provided relevant empirical information. In any case, the number of surveys, the diversity of respondents, and the recommendations and conclusions derived from the qualitative and quantitative results obtained and presented should be highlighted.

I believe that this type of study, although it has a highly sociological profile, is of interest to Land's audience and is very important in demonstrating the lack of equity and inclusion that still persists in many rural societies associated with subsistence economies. Despite its sociological character, the study ultimately addresses an issue of land uses that also have an ecological derivation that is not analyzed in this work. However, some ecological phenomena associated with land tenure are mentioned that would be interesting to address in future studies: the process of scrubbing mentioned, the more intensive use of some crops, gardening... A territorial analysis of these phenomena would be very interesting in the future.

Some detailed comments

Page 6, Line 235: “Ancestral knowledge”: What kind of knowledge? Traditional Ecological Knowledge (TEK)referred to rural activities or knowledge as wises (wisdom based on age or even mysticism or beliefs)?

Page 6, Line 248: “Life trees”: Please, explain their social/cultural/economic sense.

Page 6, Line 250: “external encroachment”: Does it refer to an advance of woody species (shrubbery) over farmland? What is the cause, the abandonment or non-exploitation of these fields?

Page 9, Lines 383 to 397: This information would be more fluid and clear if it was structured in a table or figure including the definition of each dimension (completeness...etc).

Page 11, Line 476 (Fig. 1): Please, complete and place the legend as the foot of the figure. The figures must be self-explanatory, without the need to refer to the text in order to be fully understood.

Page 12, Line 482 (Fig. 2): Please, place the legend as the foot of the figure. In this case the legend may not be completed as it stands (connecting the legend and the figure through the questions).

Page 12, Line 484: “Views on women and youth land rights did not vary much by age, but they did by gender”. Why is this information not included in the figure? It might be interesting.

Page 13, Line 519 (Fig. 3): Please, complete and place the legend as the foot of the figure.

Page 19, Line 815 (Fig. 4): Please, place the legend as the foot of the figure. In this case the legend may not be completed as it stands (connecting the legend and the figure through the questions).

Author Response

My responses to each of this reviewers comments are in italics below:

Page 6, Line 235: “Ancestral knowledge”: What kind of knowledge? Traditional Ecological Knowledge (TEK)referred to rural activities or knowledge as wises (wisdom based on age or even mysticism or beliefs)?

I removed this section because I combined the findings from the literature review with our primary findings and presented them using the framework. This information seemed less relevant. 

Page 6, Line 248: “Life trees”: Please, explain their social/cultural/economic sense.

I added an explanation for "Life trees"

Page 6, Line 250: “external encroachment”: Does it refer to an advance of woody species (shrubbery) over farmland? What is the cause, the abandonment or non-exploitation of these fields?

It refers to incursions and poaching within forests or customary land by those not considered community members. I have added an explanation  in the paper.

Page 9, Lines 383 to 397: This information would be more fluid and clear if it was structured in a table or figure including the definition of each dimension (completeness...etc).

I created a table and explained the dimensions of the land rights framework and cut down some of the narrative. This was a great suggestion, thank you! I also moved the framework description into the Materials and Methods section and added a table to explain the dimensions.

Page 11, Line 476 (Fig. 1): Please, complete and place the legend as the foot of the figure. The figures must be self-explanatory, without the need to refer to the text in order to be fully understood.

I have not addressed the comments on the figures because I did not understand them. If given more time, I can generate new charts and address these concerns.

Page 12, Line 482 (Fig. 2): Please, place the legend as the foot of the figure. In this case the legend may not be completed as it stands (connecting the legend and the figure through the questions).

I have not addressed the comments on the figures because I did not understand them. If given more time, I can generate new charts and address these concerns.

Page 12, Line 484: “Views on women and youth land rights did not vary much by age, but they did by gender”. Why is this information not included in the figure? It might be interesting.

To address this question, I would need to create a new figure. If given more time, I can generate a new charts and address these concerns.

Page 13, Line 519 (Fig. 3): Please, complete and place the legend as the foot of the figure.

I have not addressed the comments on the figures because I did not understand them. If given more time, I can generate new charts and address these concerns.

Page 19, Line 815 (Fig. 4): Please, place the legend as the foot of the figure. In this case the legend may not be completed as it stands (connecting the legend and the figure through the questions).

There is no need for a legend for this figure. I do not understand this question.

Reviewer 2 Report

The authors described the problem of land accessibility in Liberia with particular emphasis on restrictions that affect young people and women. This subject was presented on the basis of the legislation of this country and the survey of respondents. The course of research and the obtained partial results were presented in too detail. A reader from outside of Liberia may be "lost" in the sheer volume of regions and places. It would be more beneficial to present the results of the analysis grouped, e.g. by problem intensity, or other criteria. In addition, the article gives the impression of a comprehensive technical report of practical measurements. Citing international literature on the problem would give him a more scientific and methodological character. In addition to the important recommendations addressed to the Liberian authorities, the conclusions should also cover more general matters. Then the article, next to cognitive values, would also contribute to the development of research. I encourage the authors to make this extra effort, because I find the article interesting.

Author Response

Thank you for your feedback on the volume of the content being distracting. I cut out most of the general information including the long background section and focused the introduction specifically on the issues faced by youth in accessing land. I also combined the literature review with our other findings and presented them within the conceptual framework because it made sense to discuss our findings in light of other evidence on youth land rights in Liberia. This has hopefully streamlined the information in the paper.